# The Case for Clinical Trials with Novel GABAergic Drugs in Diabetes Mellitus and Obesity

**DOI:** 10.3390/life12020322

**Published:** 2022-02-21

**Authors:** Ferenc A. Antoni

**Affiliations:** Centre of Discovery Brain Sciences, Deanery of Biomedical Sciences, University of Edinburgh, Edinburgh EH8 9XD, UK; ferenc.antoni@ed.ac.uk

**Keywords:** GABA, S44819, MK-0777, GABA_A_ receptor isoforms

## Abstract

Obesity and diabetes mellitus have become the surprising menaces of relative economic well-being worldwide. Gamma amino butyric acid (GABA) has a prominent role in the control of blood glucose, energy homeostasis as well as food intake at several levels of regulation. The effects of GABA in the body are exerted through ionotropic GABA_A_ and metabotropic GABA_B_ receptors. This treatise will focus on the pharmacologic targeting of GABA_A_ receptors to reap beneficial therapeutic effects in diabetes mellitus and obesity. A new crop of drugs selectively targeting GABA_A_ receptors has been under investigation for efficacy in stroke recovery and cognitive deficits associated with schizophrenia. Although these trials have produced mixed outcomes the compounds are safe to use in humans. Preclinical evidence is summarized here to support the rationale of testing some of these compounds in diabetic patients receiving insulin in order to achieve better control of blood glucose levels and to combat the decline of cognitive performance. Potential therapeutic benefits could be achieved (i) By resetting the hypoglycemic counter-regulatory response; (ii) Through trophic actions on pancreatic islets, (iii) By the mobilization of antioxidant defence mechanisms in the brain. Furthermore, preclinical proof-of-concept work, as well as clinical trials that apply the novel GABA_A_ compounds in eating disorders, e.g., olanzapine-induced weight-gain, also appear warranted.

## 1. A Brief Primer of GABA and GABA_A_-RS

The main site of GABA production is the central nervous system (CNS) where it is an abundant neurotransmitter. The effects of GABA in the body are exerted through ionotropic (GABA_A_) and metabotropic GABA_B_ receptors [1,2,3]. The biochemical synthetic, breakdown and uptake pathways for GABA are well known [4]. Reuptake into both neurons and glial cells has been well characterized. Importantly, extensive evidence shows that glial cells can synthesize/recycle as well as release GABA to influence neuronal activity [5,6,7,8]. The concentrations of GABA in the synaptic cleft are high, ranging from 10 to 1000 µM, usually saturating and desensitizing synaptic GABA_A_ receptors (GABA_A_-R) [9]. Hence, the effects of synaptic GABA are generally thought of as “fast” and “phasic” [10]. Notably, a “slow” phasic GABAergic inhibition has been also reported [11]. Another significant category of GABA action is “tonic” inhibition, which is mediated by extra-synaptic GABA_A_-Rs [10] that are sensors for GABA in the interstitial space (“ambient GABA”) [12]. Tonic inhibition is fundamentally important in setting the firing threshold of cortical neurons [10,13]. The ambient GABA involved in this process is derived from overflow of the synaptic clefts, as well as through release from glial cells [6,7,8,14]. 

In addition to its prominent role in the CNS, potential sites of GABA action in a variety of peripheral tissues including the airways and the pancreas have been reported [15,16]. The physiological role of GABA in these systems remains to be fully explored. However, pharmacologic targeting of GABA_A_-Rs in these tissues appears to be effective in preclinical models [17,18,19,20] and offers new therapeutic opportunities. 

GABA_A_-Rs are ligand-operated Cl^-^ channels assembled from five integral membrane-protein subunits [1]. The repertory of GABA_A_-Rs subunits consists of several gene families (α, β, γ, δ, ε, θ, π, ρ) most of which have several isoforms. The most common GABA_A_-Rs configuration is 2α2βγ, a heteropentameric protein complex that spans the plasma membrane. The GABA_A_-R isoforms show distinct topographical distributions in the brain [2,21]. Moreover, molecular genetic dissection of the actions of diazepam, a non-selective positive allosteric modulator (PAM) of subunits, revealed that different GABA_A_ α-subunits mediate its plethora of pharmacological effects. At a simplified level, α1 subunits were associated with sedation and protection from epilepsy, α2 with anxiety and α5 with effects on cognitive function [22]. The role of the α3 and α4 subunits remains moot, whilst α6 is primarily expressed in the cerebellum and is thus associated with motor coordination. Subsequent studies revealed that different GABA_A_ receptor configurations underlie phasic and tonic effects of GABA [10,23,24]. Collectively, these findings provided a strong biological rationale for the development of compounds selective for GABA_A_-R isoforms [22,25,26]. 

## 2. Potential New GABA-ergic Drugs for Diabetes and Obesity

While GABA_A_ receptors are targets for a wide range of drugs such as anxiolytics, hypnotics, anaesthetics and anticonvulsants [1,27], none of the compounds currently used in the clinic is selective for a particular GABA_A_-R isoform. The more recently characterized crop of isoform-selective compounds suitable for clinical trials is listed below.

### 2.1. Benzodiazepine-Site Modulators

Previously, compounds such as diazepam or alprazolam, that enhance the effects of GABA by acting at the benzodiazepine site were referred to as agonists, while those that reduce the effect of GABA e.g., Ro15-4513 or DMCM (methyl-6,7-dimethoxy-4-ethyl-beta-carboline-3-carboxylate) were called inverse agonists. At present, the terms positive and negative allosteric modulator are preferable instead of agonist and inverse agonist, respectively. 

The main facets of the now-defunct GABA drug-development program at Merck, Sharpe and Dohme have been extensively reviewed [25,28,29]. Briefly, most of the compounds targeted the benzodiazepine allosteric modulatory site of GABA_A_-Rs. At the ligand-binding level, which is typically monitored as the displacement of ^3^H-flumazenil from high-affinity binding sites, these compounds had little if any isoform selectivity [28,29]. However, when tested in functional assays on recombinantly expressed receptors in vitro or in preclinical assays in vivo, they showed the characteristics of isoform-selective pharmacology. Thus the term “functional selectivity” was used for these molecules. Notable examples that have been tested in patients are MK-0777 (TPA-023; [30]), a positive allosteric modulator (PAM) preferring α2- and α3-GABA_A_-Rs [31] and α5IA [30,32] an α5-GABA_A_ negative allosteric modulator (NAM). While deemed essentially safe in phase 1 clinical trials, the development of both compounds was eventually stopped as MK-0777 had unfavourable bioavailability and preclinical toxicity issues [28] while a metabolite of α5IA produced kidney damage in elderly patients [28,29]. Overall, a further problem was the lack of a good correlation between preclinical findings and the effects of the drugs in patients [26,28,29]. Nonetheless, both MK-0777 and α5IA are suitable for proof of concept pilot studies in humans e.g., see [31,32]. 

Basmisanil (RG1662, RO5186582, https://en.wikipedia.org/wiki/Basmisanil, (accessed on 15 February 2022) is a NAM selective for α5-GABA_A_-R developed by Roche. This compound shows close to 50-fold selectivity for α5-GABA_A_-Rs at the ligand-binding level when compared to other GABA_A_ isoforms, thus is not just “functionally selective”. Basmisanil was safe in humans and showed clear CNS level efficacy [33]. It was tested in phase 2 clinical trials for Down’s syndrome to improve cognitive function in children (age 6–11), but the trial was terminated due to lack of efficacy https://www.downs-syndrome.org.uk/news/the-clematis-trial-conducted-by-roche/, (accessed on 15 February 2022), despite encouraging early EEG findings [34]. A phase 2 trial for improving cognitive deficits associated with schizophrenia in patients receiving non-specified anti-psychotics has also failed to meet the desired end-points https://clinicaltrials.gov/ct2/show/results/NCT02953639, (accessed on 15 February 2022).

### 2.2. Compounds Acting at the GABA-Site

Muscimol is the best-known non-selective GABA-site agonist. Isoform selective GABA_A_ agonists have not been produced so far. A priori, given the high sequence homology of the α- and β-subunits that form the GABA-binding pocket [35,36,37], it would seem unlikely that this goal is achievable. However, following on from computer-modelling studies [38], recent data indicate that a segment (loop F) of the extracellular domain of the α-subunit, that shows significant sequence variation between isoforms, determines the potency of GABA on GABA_A_ receptors [39]. In particular, these results raise the possibility of designing agonists selective for α2-GABA_A_ receptors. 

Gaboxadol (THIP, now **OV101**) is a potent agonist of GABA receptors that contain α4- or α6-subunits complemented with β- and δ-subunits, which have restricted anatomic distribution in the thalamus, hippocampus, and cerebellum and are mainly extra-synaptic in location [40,41]. This compound was also in clinical development and while it has been discontinued because of unwanted long-term side-effects, it is eminently suitable for shorter-term clinical work [42]. 

The variable segment of loop F of the α-subunit is also important in determining the isoform selectivity of novel GABA_A_ competitive antagonists such as S44819 [43,44]. S44819 (Servier) appears to be selective for extra-synaptic α5-GABA_A_-Rs [43]. Moreover, in moderate doses, this compound is effective in modifying neuronal activity in the human motor cortex [45] and has been tested in phase 2 clinical trials for post-stroke recovery. No adverse effects were detected but ultimately the drug was discontinued due to lack of efficacy [46]. This trial has been criticized due to the selection of inadequate end-points [47,48]

Taken together, at least two different benzodiazepine site compounds and a GABA-site competitive antagonist are available for clinical trials to inhibit α5-GABA_A_-Rs. In addition, MK-777 is a PAM for the simultaneous enhancement of α2- and α3-GABA_A_-Rs that is also safe to use in humans in short-term trials [31]. Gaboxadol is a clinically tested agonist for α4 and α6 subunit-containing GABA_A_ receptors. 

## 3. Diabetes Mellitus: Replenishing the Pancreatic Beta Cell Pool

### 3.1. Unmet Medical Need

Diabetes mellitus is a serious metabolic disorder that is increasingly common globally. It is a dysfunction of the endocrine islets of the pancreas caused by variable etiologies. In Type 1 diabetes the primary lesion appears to be the autoimmune destruction of the pancreatic beta-cells that produce insulin, with secondary consequences to other cell types in the islets. The essence of Type 2 diabetes is functionally inadequate insulin secretion and/or marked insulin resistance of the target organs. Whilst numerous drugs, including various formulations of the beta-cell hormone insulin, are available to combat diabetes mellitus, adequate control of plasma glucose and the long-term cardiovascular and neuropsychiatric complications of the disorder remain a serious problem. The ideal therapy would be to restore the insulin secretory capacity and physiological regulation of pancreatic islet beta cells. This remains an unmet medical need.

### 3.2. Therapeutic Rationale

Pancreatic islets are intricate organs that show major species differences with respect to the microtopography and consequent functional orchestration of the constituent cells [49,50]. The arrangement of the cells in human pancreatic islets indicates a fundamental role for paracrine interactions between alpha- and beta-cells that secrete glucagon and insulin, respectively [49]. GABA is an important paracrine factor in pancreatic islets, its levels in the pancreas are comparable to those in the brain [51,52]. All the relevant components of GABA-ergic neurotransmission, including GABA_A_ and GABA_B_ receptors, have been demonstrated in pancreatic islets [49,52,53]. Beta-cells have a considerably higher GABA content than alpha-cells [53]. A further notable aspect of GABA action in islets is that KCC2, the K^+^-Cl^−^ co-transporter protein that is required for the hyperpolarizing action of GABA in nerve cells [54] is expressed in alpha-cells but is absent from beta-cells [55]. Thus GABA depolarizes beta-cells, thereby facilitating insulin secretion and intracellular Ca^2+^ signals, while it hyperpolarizes alpha-cells and consequently inhibits the secretion of glucagon [52]. More significantly, it was shown by several groups that chronic administration of GABA in streptozotocin-diabetic mice converts islet alpha-cells into beta-cell-like insulin-producing cells and normalizes plasma glucose levels [19,56,57,58]. Treatment with GABAergic compounds also improves the survival and proliferation of beta cells of human islets grafted into mice [19,56,57,58]. Several of the GABA_A_-R subunits have been found in human pancreatic islets [52,59,60,61], and the localization of the α2-subunit in beta-cells appears particularly consistent and potentially significant [60,62]. 

Since the discovery that the GABA biosynthetic enzyme GAD65 is a target antigen in autoimmune diabetes [63,64], interest in the role of GABA in the immune system has intensified. Various immunomodulatory actions of GABA_A_ receptors have been reported [20], which could be significant with respect to the autoimmune destruction of islet beta cells as well as the role of inflammation in insulin resistance in Type 2 diabetics. Evidence for the expression of several GABA_A_ receptor subunits by immunocompetent cells has been provided at the mRNA level [20,65,66]. However, it is ultimately unclear which receptor configurations mediate the purported anti-inflammatory actions of GABA [20,67,68]. At this point, expression of the α1-subunit appears to be most frequently reported. Importantly, evidence for possible inflammatory effects of GABA has also been generated [69]. 

### 3.3. Potential Mode of Pharmacologic Intervention 

The data outlined above indicate that activation of α2-GABA_A_ receptors could prove beneficial in improving the deficiency of islet beta-cell function. Although GABA is available as a food supplement, it is unclear whether or not it is non-toxic at the dosage required for an islet-trophic effect. A recent report on the pharmacokinetics and tolerability of high doses of GABA has been published with no major adverse effects of 2 g of GABA three times daily for seven days. However, a recent preclinical study reported the appearance of nonalcoholic steatohepatitis (NASH) in mice fed a methionine-choline deficient diet and dosed with 2 mg/mL GABA in the drinking water for four weeks [69]. Moreover, inflammatory actions of GABA mediated by bicuculline-sensitive GABA receptors were also found in the same study. Further analysis is required to assess the significance of these findings with respect to the use of GABA as an islet-trophic agent. An option to reduce the dose of GABA required for islet-trophic effects would be to co-administer a GABA_A_ PAM [19], preferably one that does not penetrate the CNS to avoid unwanted side effects. Most likely, GABA_A_ PAMs that do not penetrate the blood-brain barrier already exist, except that until recently they were not thought to be of therapeutic value. However, there is a weakness to this approach as the unwanted side-effects of GABA could be also enhanced by a PAM that is not GABA_A_ isoform-selective. The optimal choice appears to be the combination of an α2-GABA_A_-R selective PAM with a CNS-penetrating α5 inhibitor (NAM or GABA antagonist). This combination would enhance GABA action at the islets, potentially ameliorate the dysfunction of the hypoglycemia counter-regulatory response at the VMN and diminish the probability of cognitive impairment. The current crop of compounds, MK-0777 and α5IA or S44819 would be suitable for a clinical trial along this therapeutic principle. As a caveat, it should be mentioned that in vitro MK-0777 is substantially more potent on α3-GABA_A_ receptors than on α2 containing ones [70,71]. Finally, experimental compounds that selectively potentiate α2-GABA_A_ receptors and inhibit α5-GABA_A_ containing ones are worth mentioning here [72]. 

## 4. Diabetes Mellitus: Prevention of Iatrogenic Hypoglycemia in Insulin-Dependent Diabetics

### 4.1. Unmet Medical Need

Recurrent iatrogenic hypoglycemia leading to hypoglycaemia associated autonomic failure is a serious complication in insulin-treated diabetes and is a limiting factor in maintaining proper glycemic control [73,74,75]. Repeated hypoglycaemia induced by insulin attenuates the endocrine counter-regulatory response to hypoglycemia, which normally involves a marked increase of plasma glucagon, norepinephrine and cortisol [73,74]. As a result, patients receiving chronic insulin therapy undergo episodes of poorly compensated hypoglycaemia and in many cases develop impaired perception of hypoglycemia. Hypoglycemia can lead to neuronal degeneration [76], and moreover diabetes facilitates brain small vessel disease, thereby potentially further aggravating the damage to neuronal function [77]. Currently, there are no adequate measures to prevent this complication of insulin therapy. It is of note, that the attenuation of the hypoglycemic counter-regulatory response can be also induced by repeated hypoglycemia in rodents as well as non-diabetic human subjects [78]. 

### 4.2. Therapeutic Rationale 

The brain, and particularly the ventromedial nucleus of the hypothalamus (VMN), plays a crucial role in sensing hypoglycemia and initiating the physiological counter-regulatory responses that rapidly correct it [73,79,80,81]. There is a large body of preclinical evidence implicating GABAergic neurotransmission in the VMN in the etiopathology of recurrent iatrogenic hypoglycemia induced by insulin therapy [73,80]. Significantly, repeated hypoglycemia markedly increases the levels of the GABA synthetic enzyme GAD as well as GABA in the ventromedial hypothalamus of rats [82]. Moreover, in rats with an attenuated counter-regulatory response induced by repeated hypoglycemia, localized inhibition of ionotropic GABA (GABA_A_) receptors in the ventromedial hypothalamus restored the endocrine counter-regulatory response [82]. 

The anatomical topography of GABAergic neurotransmission in the VMN is remarkable in that the VMN contains virtually no cell bodies that express the key GABA-synthetic enzymes GAD 65 or 67 [83]. Thus, GABA in the VMN is largely derived from afferent nerve fibres originating outside of the nucleus. With respect to the potential sources of the GABAergic input to the VMN, the dorsomedial hypothalamic nucleus, the lateral hypothalamic area, the arcuate nucleus, the tuberomamillary nucleus and the amygdala are most notable [84]. In the case of the tuberomamillary nucleus, the GABAergic afferents also contain and release histamine [85]. With respect to the distribution of GABA_A_-R, the dorsomedial part of the VMN stands out as expressing high levels of α2 and α5 subunits, whereas the α3 subunit appears less abundant and tends to be localized in the ventrolateral aspect of the VMN [21,86,87,88,89]. By comparison, the expression of GABA_A_-R α-subunits in the arcuate nucleus, which is an important hypothalamic centre of nutrition and food intake, appeared to be much lower and consisted mainly of the α1 isoform [21,88]. 

### 4.3. Potential Mode of Pharmacologic Intervention 

From a pharmacological point of view, it would be relevant to examine whether or not treatment with α5IA or S44819 would reverse the insulin-induced reprogramming of the counter-regulatory response to hypoglycemia or even prevent it from developing in the first place. Should these studies yield negative results further work with MK-0777 or the combination of MK-0777 and one of the α5-GABA_A_-R inhibitors appears warranted. 

## 5. Diabetes Mellitus: Combating Cognitive Decline

### 5.1. Unmet Medical Need 

An important long-term side-effect of insulin-dependent diabetes is the impairment of cognitive function, particularly attention and cognitive flexibility [90]. One causative factor in this process is likely to be recurrent hypoglycemia, especially if the subjects have impaired awareness of it [91]. In the case of type 2 diabetes, patients have been reported to have increased brain concentrations of GABA, which was correlated with the degree of the impairment of cognitive function observed in this condition [92]. In sum, both types of diabetes are associated with a slow decline in cognitive performance despite efforts to achieve glycemic control. Cognitive impairment is a serious impediment to self-sufficiency and thus an increasing social burden in the ageing human population. Taken together, adjuvant procognitive therapy to attenuate the decline of cognitive performance in diabetic patients is an unmet medical need. 

### 5.2. Therapeutic Rationale

The mechanisms underlying the neurotoxic effect of hypoglycemic episodes in insulin-dependent diabetes are thought to be similar to those involved in stroke—hypoperfusion-reperfusion injury [76] a major element of which is oxidative damage [93]. Preclinical studies have demonstrated that inhibition of α-5 GABA_A_Rs ameliorates the impairment of cognitive performance induced by various etiologies [94,95,96,97,98] including stroke [99]. Significantly, these compounds also enhanced the post-stroke recovery of sensory-motor function in rodent models [99,100]. In a more recent study, S44819 was shown to prevent/reverse the impairment of working and recognition memory induced by mild stroke in rats or chronic cerebral hypoperfusion in mice [98]. The likely mechanisms underlying these effects of α-5 GABA_A_ inhibition include facilitation of synaptic plasticity [29,43] and the enhancement of spontaneous neuronal network activity that favours antioxidant neuroprotective processes in the brain [98,101,102].

### 5.3. Potential Mode of Pharmacologic Intervention 

Inhibition of α5-GABA_A_Rs (α5IA, basmisanil or S44819) as a therapy to improve cognitive function in diabetes appears to be worthwhile, because of the anticipated neuroprotective effects and the remediation of the dysfunction of synaptic plasticity. Moreover, the same agents may also ameliorate the defect in the counter-regulatory response to hypoglycemia as outlined in the previous section and thereby improve glycemic control. Last, but not least, α5-GABA_A_R inhibition could also prove beneficial with respect to comorbidities such as anxiety and depression [103] that are often associated with type 1 diabetes. With respect to Type 2 diabetes, the recent clinical findings indicating an increase of brain GABA levels and their potential impact on cognitive function also justify the therapeutic concept of α5-GABA_A_-R inhibition. A summary of the proposed adjunct GABAergic pharmacologic therapy of insulin-dependent diabetes mellitus is shown in Figure 1.

## 6. Obesity: The Case of Olanzapine Induced Weight Gain 

### 6.1. Unmet Medical Need 

Olanzapine (Zyprexa) is one of the most frequently prescribed antipsychotics. As with several other antipsychotics in this class, it is known to increase body weight, sometimes leading to metabolic syndrome and Type 2 diabetes [104]. As olanzapine and atypical antipsychotics are valuable for several therapeutic applications that require chronic administration, the control of olanzapine-induced body weight gain is an unmet medical need. 

### 6.2. Principle of Therapeutic Intervention

Despite numerous studies addressing the underlying causes of olanzapine-induced weight gain, there is no consensus on the subject [105]. An important problem is the lack of good preclinical models: female rats readily gain weight on olanzapine, but males do not gain sufficient weight on a normal diet to achieve statistical power with realistic numbers of experimental animals [106]. Functional studies point to the role of the antagonism of histamine 1 (H1) receptors [107] and 5-HT2c receptors [108] by olanzapine. Hypothalamic AMP kinase activity is enhanced by H1 antagonists and provides a parsimonious explanation of weight gain [107]. However, there are studies that fail to confirm this [109]. A direct metabolic effect of olanzapine on cellular nutrient handling, independent of its antipsychotic receptor profile, has been also proposed [110]. 

Whilst food-intake studies in rodents have notoriously low translational potential, it is interesting to highlight here certain parallels between olanzapine-induced weight gain in humans and that induced by lesioning of the VMN in rats (see [111,112] for review). Both treatments produce a transient weight gain as if attaining a new set-point [113]. Both work most reliably in females [111,114] and disrupt the normal secretion of adrenal corticosteroids [115,116]. A recent study highlighted a potential sexually dimorphic mechanism by which metabolic GluR5 receptors interact with estrogen receptors to alter the function of steroidogenic factor 1 (SF1) positive VMN neurons [117], i.e., indicating the potential for female-specific regulation of metabolism at the level of the VMN. The SF1 neurons are known to regulate the sensitivity to insulin in peripheral tissues [118] as well as to suppress food intake through projections to the paraventricular thalamic nuclei [119]. In a yet further aspect, histamine H1 receptors, implicated in the weight-gaining effects of olanzapine in humans, are prominent in the VMN and histaminergic activation of the VMN in rats reduces food intake [120,121,122]. On the basis of these findings, it is attractive to speculate that a major action of olanzapine is to impair the anorexic functionality of the VMN. As the VMN prominently expresses α2- and α5-GABA_A_ receptors, and in view of the reported interaction between the actions of histamine and GABA in the control of the excitability of VMN neurons [123] further work exploring the role of these receptors in olanzapine-induced weight gain appears justified.

As the first human genetic approach to the problem, a recent genome-wide association study was targeted at weight gain induced by the atypical antipsychotics clozapine or olanzapine [124]. A highly significant association of antipsychotic-induced weight gain with polymorphisms in the GABRA2 gene that encodes the GABA_A_-R α2-subunit was found. Previous genome-wide association studies have implicated GABRA2 in the obesity of normal subjects [125]. 

### 6.3. Potential Mode of Pharmacologic Intervention

The findings outlined above prompted us to examine the effect of the α2/α3-GABA_A_ PAM, MK-0777 on olanzapine-induced weight gain in female Sprague–Dawley rats. Interestingly, MK-0777 (10 mg/kg twice daily) failed to prevent the weight gain induced by olanzapine administration. However, once the typical plateau of olanzapine-induced weight gain was reached, a significant degree of weight loss was observed so that body weight gain after two weeks of simultaneous treatment with olanzapine and MK-0777 was not different from that in the vehicle-treated group (Figure 2A). Under identical conditions the α5-GABA_A_ NAM α5IA had no significant effect on olanzapine induced weight gain (Figure 2B). 

Thus, further exploration of the effects of MK-0777 or further α2/α3-GABA_A_ PAMs on olanzapine-induced weight gain appears warranted. An adjuvant therapy achieving α2-GABA enhancement as well as α5-GABA inhibition could be also advantageous, as olanzapine-induced weight gain, and disturbances of plasma glucose regulation, as well as cognitive deficits associated with schizophrenia [126] may be all improved. 

## 7. Future Perspectives

A major advance for GABAergic therapeutics would be a more detailed structural understanding of the way in which drugs interact with the heteropentameric complex [129]. Cryo-electronmicroscopy should help in this respect [130], but will have to be aided by special pharmacological agents that allow exploration of the role played by, for instance, loop F of the extracellular domain in ligand recognition [39,131] and various other sites of drug-receptor interaction [132]. This approach may well lead to selective α2-GABA_A_ agonists that could significantly improve the therapy of diabetes mellitus. Moreover, such compounds could also have favourable properties for the control of stress, anxiety and overeating [133]. In addition to the supply of novel drugs, studies of human genetic polymorphisms and epigenetics are likely to yield important new applications of existing GABAergic compounds. 

*Footnote:* + Note that it in some publications, ventromedial hypothalamus and ventromedial nucleus (VMN) are used interchangeably. However, in addition to the VMN, the ventromedial hypothalamus also contains the arcuate nucleus and the median eminence with their respective afferent inputs.

## Figures and Tables

**Figure 1 life-12-00322-f001:**
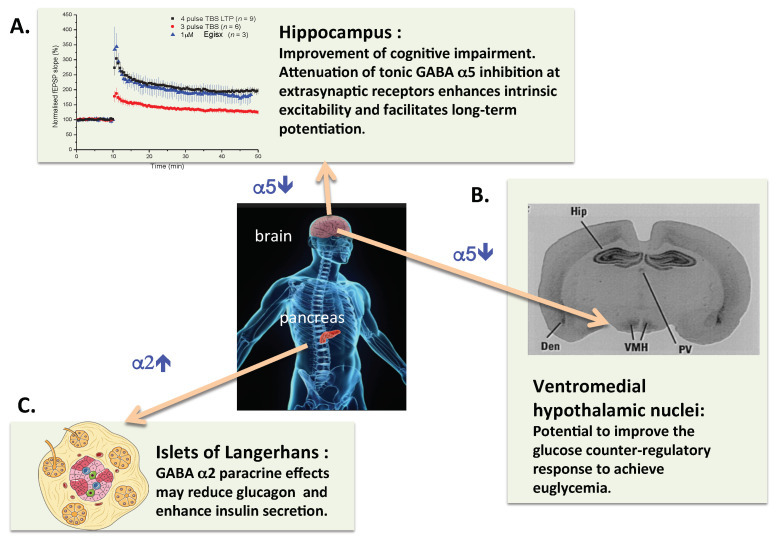
Schematic representation of the potential therapeutic benefits of treatment of insulin-dependent diabetic patients with a combination of an α2-GABA_A_ enhancer and an α5-GABA_A_ inhibitor compound. (**A**) It is well established in preclinical models that inhibitors of α5-GABA_A_-R facilitate hippocampal synaptic plasticity and can improve impairment of learning and memory. (**B**) In addition to the hippocampus, the hypothalamic ventromedial nuclei stand out with high levels of α5-GABA_A_ binding sites in rat brain [86]. This has been confirmed by mRNA in situ hybridization in human brain. Thus it is justified to hypothesize that inhibition of α5-GABA_A_-R could reduce the GABAergic impairment of the glucose counterregulatory response in insulin-dependent diabetics. (**C**) The insulinotrophic effects of GABA appear to be mediated by α2-GABA_A_-R, thus an α2-GABA_A_ PAM could improve pancreatic islet beta cell function. Finally, it is of note that compounds with dual, α2-GABA_A_ enhancer and α5-GABA_A_ inhibitor activity have been produced [72].

**Figure 2 life-12-00322-f002:**
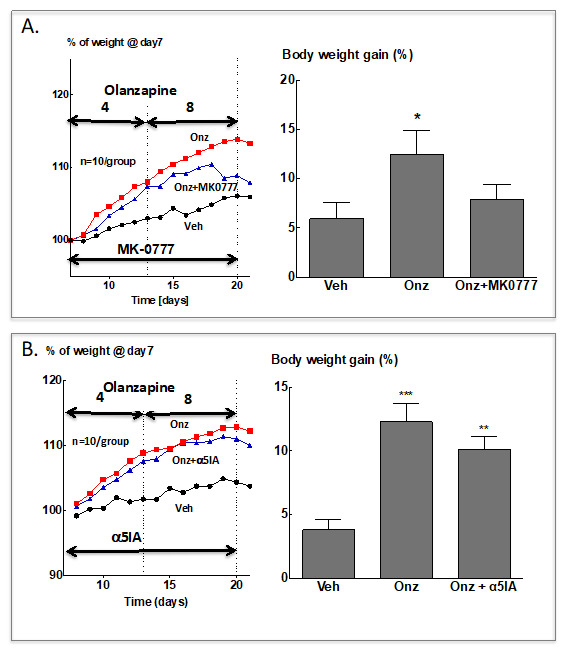
Effect of treatment with GABA_A_-R modulators on olanzapine-induced increases of body weight in female Sprague–Dawley rats (G. Gigler, I. Gacsályi, F. A. Antoni, unpublished data). Female rats were allowed to accommodate in individual cages and were weighed daily for 7 days. Vehicle (0.9% saline *w*/*v*) and olanzapine treatment were started on day 8, 4 mg/kg *p.o.* per day for 7 days and subsequently 8 mg/kg for the next 7 days [127]. (**A**) MK-0777 was suspended in 1% Tween 80 and given at 10 mg/kg *i.p.* at the same time (2 h before lights off) as olanzapine; vehicle and olanzapine only treated animals received 1% Tween 80 (2 mL/kg) *i.p. Left panel:* Means, error bars are not shown for the sake of clarity. *Right panel:* Mean body weights on day 21 after the last administration of drugs. Means ± S.E.M. are shown, N = 10/group. Data were analysed by one-way ANOVA followed by Dunnett’s test for multiple comparisons. * *p* < 0.05 vs. the vehicle group. (**B**) As in A. except that α5IA [128] was the GABA_A_-R modulator used. ** *p* < 0.01, *** *p* < 0.001 vs. the vehicle group.

## Data Availability

Not applicable.

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
