# Peer review of "The Case for Clinical Trials with Novel GABAergic Drugs in Diabetes Mellitus and Obesity"

_life, 2022, doi:10.3390/life12020322_

Round 1

Reviewer 1 Report

Points of criticism:

The author should give a summary conclusion on each chapter in the paper. The text now gives reasons for Unmet medical need and then presents a diverse report of results from different papers but leaves the reader to themselves make some kind of conclusion. I think it would be helpful for the reader to read what conclusion the author draws from the described results.

The authors discuss the alpha 2 and alpha 5 receptor subtypes but the most abundant subtypes in hypothalamus are alpha 1, 2, 3 and 5 (Wisden et al., 1992) with β-subunits 1, 2 and 3, and γ-subunits 2 and 3 (Pirker et al., 2000, Backberg et al., 2004). Studies of benzodiazepine-induced food intake have suggested that α3 and α2 GABAA receptor α-subunits are involved, but not α1-, or α5-subunit containing receptors (Cooper, 2005). Morris et al. (2009) have found that α3-containing receptors specifically mediate benzodiazepine-induced hyperphagia in rodents. However, the alpha3 subunit is not discussed properly in the paper this should be discussed more. The result of benzodiazepine and alcohol induced overeating and obesity is not discussed properly and should be expanded in the paper.

There are several abbreviations in the paper, and it would be good to have a table with abbreviations in the beginning of the paper.

Explain the abbreviation:

VMN wright out the first time the abbreviations are mentioned

VMH is this a spelling error? If not wright out the first time the abbreviations are mentioned

In Figure 1 the abbreviation alpha5IA needs an explanation in the figure legend.

Author Response

Reviewer 1

I would to like to thank Reviewer 1 for valuable comments on the manuscript.

“The author should give a summary conclusion on each chapter in the paper.”

I presume chapters are the sections. I have inserted a summary sentence at the end of the section outlining GABAa receptor pharmacology, which also introduces the relatively selective GABA-ergic compounds that are currently available clinical trials.

“The text now gives reasons for Unmet medical need and then presents a diverse report of results from different papers but leaves the reader to themselves make some kind of conclusion. I think it would be helpful for the reader to read what conclusion the author draws from the described results.”

The remaining sections:

DIABETES MELLITUS: REPLENISHING THE PANCREATIC BETA CELL POOL

DIABETES MELLITUS: PREVENTION OF IATROGENIC HYPOGLYCEMIA IN INSULIN-DEPENDENT DIABETICS

DIABETES MELLITUS: COMBATING COGNITIVE DECLINE

OBESITY: THE CASE OF OLANZAPINE INDUCED WEIGHT GAIN

are structured uniformly: 1) unmet medical need 2) therapeutic rationale 3) the proposed mode of pharmacological intervention with the drugs introduced in the first section. These are very specific and require no further concluding statements.

“The authors discuss the alpha 2 and alpha 5 receptor subtypes but the most abundant subtypes in hypothalamus are alpha 1, 2, 3 and 5 (Wisden et al., 1992) with β-subunits 1, 2 and 3, and γ-subunits 2 and 3 (Pirker et al., 2000, Backberg et al., 2004). “

The current crop of GABAa receptor drugs was designed to target the alpha subunits of the receptor as these are associated with distinct biological functions – as clearly set out in the first section of the paper. There are some ideas as to which beta and gamma subunits preferentially associate with a particular alpha subunit but there is no evidence that this fundamentally alters the pharmacological properties of the receptor. Pirker et al 2000 is ref22 Bäckberg et al 2004 is ref 88 in the bibliography. Precisely these studies showed the unusually high level of expression of alpha 5 GABAA subunits in the ventromedial hypothalamic nuclei when compared with other alpha subunits.

"Studies of benzodiazepine-induced food intake have suggested that α3 and α2 GABAA receptor α-subunits are involved, but not α1-, or α5-subunit containing receptors (Cooper, 2005). Morris et al. (2009) have found that α3-containing receptors specifically mediate benzodiazepine-induced hyperphagia in rodents. However, the alpha3 subunit is not discussed properly in the paper this should be discussed more. The result of benzodiazepine and alcohol induced overeating and obesity is not discussed properly and should be expanded in the paper."

This paper is not about benzodiazepine or alcohol-induced feeding, Thus, I don't see the relevance of this comment by the Reviewer 1. None of the drugs that are clinically approved target the alpha3 subunit selectively. Furthermore, a key problem with the application of the benzodiazepine-site drugs in man was that the preclinical work was only weakly predictive. The therapeutic approaches I have suggested in this paper are supported by human brain anatomy and the analysis of human pancreatic beta cells.

“VMN wright out the first time the abbreviations are mentioned”

This can be found on page 11 of the original manuscript the first line of “Therapeutic rationale”

“VMH is this a spelling error? If not wright out the first time the abbreviations are mentioned”

Thank you for highlighting this, ultimately, it should be VMN. However, some authors use VMH and it is not always clear what they mean by it.

“In Figure 1 the abbreviation alpha5IA needs an explanation in the figure legend”

This is not an abbreviation it is the name of the compound e.g. see

Atack, J.R. “Preclinical and clinical pharmacology of the GABAA receptor alpha5 subtype-selective inverse agonist alpha5IA.” Pharmacol Ther 2010. 125, 11-26. I have referenced this paper in the figure legend.

Reviewer 2 Report

A fluently written and well readable review on a scientifically very relevant topic, supported by some pilot studies performed by the author himself. However, I have some remarks / suggestions / questions addressing of which may improve the final version of the manuscript:

- According to some sources, the clinical development of TPA-023 (MK-0777) was dropped because of preclinical toxicity (cataract) in long-term dosing studies. You, however, mention "unfavourable kinetic properties" as the reason - could you expand on that? Another related question: you mention that gaboxadol and MK-0777 are suitable for short-term clinical work and proof-of-concept studies. How long is "short-term" in this context and would it be enough to see an effect on body weight and/or cognitive decline?

- the order of citations in brackets - it would be more readable to put them in order: (27;30;31) rather than (31;27;30).

- Figure 1A - how do explain a sudden drop in body weight olanzapine/MK-0777 group on Day 19 - could you propose any hypotheses for the mechanism for such an effect or are there any other possible reasons other than the modulatory effect of MK-0777?

- as for Basmisanil, at least one of the studies on the effect of this compound on cognitive deficits in SCZ has already been completed: https://clinicaltrials.gov/ct2/show/results/NCT02953639 - an update with a short overview of the effects would be welcome

- Lines 125-127: "was discontinued due to lack of efficacy (47) but see (48)."I find the text unnecessarily concentrated here (and in a couple of other places) - it would be nice to add a sentence on (48) so that the reader of the review gets the essential point without needing to go back to the original article 48.

- Rajagopal et al (2018) "TPA-023 attenuates subchronic..." Neuropsychopharmacology - I feel that this is in one of the more recent  studies presenting substantial evidence that is missing from the review

Author Response

Reviewer 2

"A fluently written and well readable review on a scientifically very relevant topic, supported by some pilot studies performed by the author himself. However, I have some remarks / suggestions / questions addressing of which may improve the final version of the manuscript:"

I was gratified by the favourable comments of Reviewer 3.

-" According to some sources, the clinical development of TPA-023 (MK-0777) was dropped because of preclinical toxicity (cataract) in long-term dosing studies. You, however, mention "unfavourable kinetic properties" as the reason - could you expand on that?"

The bioavailability of MK077 (aka TPA023) was 1% as per a review by John Atack (ref 2 in paper ). Possibly formulation could have improved on that but the preclinical toxicity issue became paramount. I have modified the text accordingly.

“Another related question: you mention that gaboxadol and MK-0777 are suitable for short-term clinical work and proof-of-concept studies. How long is "short-term" in this context and would it be enough to see an effect on body weight and/or cognitive decline?”

All of the compounds mentioned in the opaper are suitable for clinical studies. One would have to request the respective updated clinical investigator brochures to give a definitive answer to this question. I would expect a cognitive improvement effect of Alpha5 inhibitors in mild cognitive impairment to be acute i.e. evident within a single trial. The key is to choose the right trial(s). MK0777 was used for over a month in a clinical study.

- the order of citations in brackets - it would be more readable to put them in order: (27;30;31) rather than (31;27;30).

Rectified.

- Figure 1A - how do explain a sudden drop in body weight olanzapine/MK-0777 group on Day 19 - could you propose any hypotheses for the mechanism for such an effect or are there any other possible reasons other than the modulatory effect of MK-0777?

The weight gain of the Olanzapine + MK0777 group stalled shortly after the olanzapine dose was switched to 8mg/kg and declined (along with food intake) on the last 3 days. This could mean that the body-weight set-point dictated by the higher dose of olanzapine was blocked by the drug. The purpose of including these data is because some human parallels are apparent in the literature and possibly others may revisit the issue.

"- as for Basmisanil, at least one of the studies on the effect of this compound on cognitive deficits in SCZ has already been completed: https://clinicaltrials.gov/ct2/show/results/NCT02953639 - an update with a short overview of the effects would be welcome"

A comment on of this (unsuccessful) trial is included.

- Lines 125-127: "was discontinued due to lack of efficacy (47) but see (48)."I find the text unnecessarily concentrated here (and in a couple of other places) - it would be nice to add a sentence on (48) so that the reader of the review gets the essential point without needing to go back to the original article 48.

Text expanded to provide further brief details. The riposte – or rather the lack of it – to ref 48 is also included.

- Rajagopal et al (2018) "TPA-023 attenuates subchronic..." Neuropsychopharmacology - I feel that this is in one of the more recent studies presenting substantial evidence that is missing from the review

Thank you for highlighting this paper. There were two previous human studies that failed to show a benefit of MK0777 on cognitive function in schizophrenia. To my mind this is a preclinical study to polish lurasidone, which has failed to live up to claims that it would restore cognitive function towards normal in schizophrenic patients.  

Reviewer 3 Report

The manuscript entitled "The case for clinical trials with novel GABAergic drugs in diabetes mellitus and obesity" is a well-presented manuscript. However, the following points need to be justified before considering them for publication. 

  1. The oxidative stress can be linked with the parameters presented in this manuscript.
  2. A schematic representation of how these GABAergic drugs work on DM and Obesity would be better
  3. The references need to be formatted and keep the updated references. 
  4. The categorization of the manuscript is very well. 
  5. Any ethical concerns related to this manuscript?

Author Response

Reviewer 3

The manuscript entitled "The case for clinical trials with novel GABAergic drugs in diabetes mellitus and obesity" is a well-presented manuscript. However, the following points need to be justified before considering them for publication. 

 I was gratified by the favourable comments of Reviewer 3.

  1. The oxidative stress can be linked with the parameters presented in this manuscript.

It is unclear what Reviewer 3 had in mind here, other than the issues concerning repeated hypoglycemia.

  1. A schematic representation of how these GABAergic drugs work on DM and Obesity would be better  

A new figure 1 is supplied

  1. The references need to be formatted and keep the updated references. 

The formatting has been checked and updated

  1. The categorization of the manuscript is very well. 

Thank you for your supportive comment.

  1. Any ethical concerns related to this manuscript?

Ethical authorization for the included animal work is quoted in the legend to Figure 2.

Round 2

Reviewer 1 Report

Paper now OK